# Preeclampsia Is a Syndrome with a Cascade of Pathophysiologic Events

**DOI:** 10.3390/jcm9072245

**Published:** 2020-07-15

**Authors:** Wilfried Gyselaers

**Affiliations:** 1Department Obstetrics, Ziekenhuis Oost Limburg, B3600 Genk, Belgium; wilfried.gyselaers@uhasselt.be; Tel.: +32-89-306420; 2Department Physiology, Hasselt University, B3590 Diepenbeek, Belgium

**Keywords:** preeclampsia, gestational physiology, pathophysiology, inflammation, immune tolerance, extracellular vesicles, maternal hemodynamics, venous hemodynamics, venous congestion, intra-abdominal pressure

## Abstract

This review integrates the currently available information on the molecular, cellular, and systemic mechanisms involved in the pathophysiology of preeclampsia. It highlights that the growth, protection, and promotion of the conceptus requires the modulation of an intact maternal immune system, communication between the mother and fetus, and adaptation of the maternal organic functions. A malfunction in any of these factors, on either side, will result in a failure of the cascade of events required for the normal course of pregnancy. Maladaptive processes, initially aiming to protect the conceptus, fail to anticipate the gradually increasing cardiovascular volume load during the course of pregnancy. As a result, multiple organ dysfunctions install progressively and eventually reach a state where mother and/or fetus are at risk of severe morbidity or even mortality, and where the termination of pregnancy becomes the least harmful solution. The helicopter view on pathophysiologic processes associated with preeclampsia, as presented in this paper, illustrates that the etiology of preeclampsia cannot be reduced to one single mechanism, but is to be considered a cascade of consecutive events, fundamentally not unique to pregnancy.

## 1. Introduction

Gestational hypertensive disorders (GHDs) account for an important fraction of prenatal complications, and maternal and perinatal near miss or mortality [1]. GHDs are a heterogeneous group of syndromes, in which elevated blood pressure is the predominant feature, occurring either before midgestation or developing during the second half of pregnancy. Table 1 lists the diagnostic criteria for different types of GHD, as currently defined by the American College of Obstetricians and Gynecologists (ACOG) and the International Society for the Study of Hypertension in Pregnancy (ISSHP) [2,3], together with the subcategories considered by different international societies [4]. 

For ages, numerous hypotheses on the etiology and pathophysiology of preeclampsia have been reported, without a generally accepted consensus today. This narrative review aims to combine currently published evidence on the sequence of background mechanisms from periconception to the full clinical syndrome preeclampsia, and put these into perspective with the known pathophysiologic processes of systemic syndromes in non-pregnant individuals as cardiorenal syndrome and other. For this, an extended literature search in PubMed was conducted using combinations of the key words: preeclampsia, gestational hypertensive disorders, epidemiology, risk factors, classification, early onset preeclampsia, placental preeclampsia, late onset preeclampsia, maternal preeclampsia, spiral artery, spiral artery remodeling, gestational adaptation, gestational physiology, intervillous space, immune tolerance, inflammation, oxidative stress, systemic inflammatory response syndrome, epigenetics, microRNA, syncytiotrophoblast extracellular vesicles, trophoblast plugs, uterine arteriovenous anastomoses, nitric oxide, maternal hemodynamics, venous hemodynamics, venous congestion, intraabdominal pressure, abdominal hypertension, abdominal compartment syndrome.

### 1.1. Epidemiology of Preeclampsia

It is estimated that GHDs occur in 5.2–8.2% of all pregnancies [5]. Overall, nonproteinuric gestational hypertension (GH) occurs in 1.8–4.4% and preeclampsia (PE) in 0.2–9.2% of pregnancies. Important differences have been reported in the rates of PE between continents: 0.5–2.3% in Africa, 0.2–6.7% in Asia, 2.8–9.2% in Oceania, 2.8–5.2% in Europe, 2.6–4.0% in North America, and 1.8–7.7% in South America [5]. Increasing rates of PE have been reported in the USA and Norway, with age-adjusted values increasing from 2.4% in 1987–1988 to 2.9% in 2003–2004 and from 3.7% in 1988–1992 to 4.4% in 1998–2002, respectively [6]. The prevalence of PE seems subject to meteorological and geographic influences because higher rates are reported in winter births and in the northern regions of Finland rather than the southern regions, and also during the transition from the dry to the rainy season in Zimbabwe [6].

### 1.2. Predisposing Risk Factors

Numerous risk factors for the development of PE have been identified. Epidemiological studies have demonstrated a genetic influence, via both maternal and paternal inheritance, with evidence for familial clustering, parental transfer, twin sibling susceptibility, and racial differences [7]. First-degree relatives of affected individuals have a 3–5-fold increased risk of PE [8]. Pregnancies fathered by partners who themselves were born after PE are twice as likely to develop PE, and men who have already conceived a preeclamptic pregnancy are at greater risk of recurrent PE in subsequent pregnancies [8,9]. In mono- and dizygotic female twin pairs, PE heritability is estimated to be 54%. A meta-analysis of individual gene studies showed that individual genetic variants encoding components of the renin–angiotensin system, coagulation, fibrinolysis, lipid metabolism, and inflammation pose an increased risk of PE, and many of these variants are also associated with an increased risk of cardiovascular disease [8,10]. Compared with White women, the risk of developing PE is higher in Black women and lower in Hispanics, Asians, and Pacific islanders [11]. Inter-racial differences in the prevalence of comorbidities, such as obesity and diabetes, may partly explain these observations, but ethnic variants of single-nucleotide polymorphisms have also been identified, such as in the endoplasmic reticulum aminopeptidase 2 (ERAP2) gene [7,12].

Primigravid women show a three-fold higher rate of PE than multigravida women, and those who have had limited reproductive contact with paternal antigens in the preconception period are at particular risk. This circumstance can arise from a short prepregnancy period of cohabitation, the use of a barrier method of contraception, a new partnership, azoospermia treated with surgical sperm extraction before intracytoplasmic sperm injection (ICSI), or donor insemination [6,9]. These observations strongly suggest that the maternal immune system plays an important role in the etiology of PE, as will be discussed further in the section on the immune system, implantation, and vascular remodeling.

Maternal comorbidities may strongly predispose a woman to GHD. A meta-analysis of large cohort studies and a World Health Organization (WHO) multicountry survey of maternal and newborn health showed that women with antiphospholipid syndrome or chronic hypertension have the highest individual risk of PE, followed by those with pregestational diabetes, chronic diseases of the kidney, liver, or heart, anemia, systemic lupus, or other systemic infections [13,14]. However, the relevance of these disorders to population health care differs markedly from their relevance to an individual’s risk and depends upon the prevalence of these morbidities in the population [13].

Another important group of predisposing factors for PE are individual characteristics, such as body mass index (BMI) > 25 kg/m^2^, age > 35 years, school education ≤ 10 years, grand multiparity (≥5), and a history of PE, abruption, or stillbirth [14]. A predisposition to gestation-specific PE is also observed in multiple pregnancies [14], and after conception with artificial reproductive treatments, particularly in vitro fertilization and ICSI techniques [15], irrespective of whether the gametes are from partners or donors or whether it is a singleton or multiple pregnancy [9,14,16,17].

Environmental factors, such as air pollution, are becoming a more and more important predisposing factor for preeclampsia and cardiovascular disease, most likely via oxidative stress induced endothelial dysfunction [18]. This mechanism will be explained further in the chapter on Immune system, implantation, and vascular remodeling.

Less-frequently reported factors associated with a higher risk for PE are maternal dyslipidemia, sleep-disordered breathing, (worsening) migraine, and the intake of certain foods, minerals, and vitamins [18,19,20,21,22,23]. It is unclear whether these factors act independently or as confounders of an underlying predisposition to cardiovascular disease [24].

## 2. Pathophysiology

### 2.1. Types of PE

The term “eklampsia” (violent bursts, lightning) was first used in ancient Greece to describe the unexpected and abrupt appearance of seizures in young pregnant women [25]. Only by the end of the 19th century to the early 20th century were hypertension and proteinuria recognized as preceding symptoms [26]. Decades later, it was noted that this disease is most prevalent in primigravid women. In the 1970s, histological studies of placental invasion were reported, and maladaptation of the maternal spiral arteries in a very early stage of pregnancy was linked to the development of PE. By the end of the 20th century, the focus of research had moved to the immunological mechanisms of systemic inflammation and endothelial activation [26]. More recently, there has been an increasing number of publications on the subtypes of PE, with different predisposing risk factors and preclinical pathophysiological pathways. The first classification system discriminated early-onset from late-onset PE based on whether the clinical diagnosis was made before or after 34 weeks [27]. These two types showed differences in maternal cardiovascular function in the latent phase of the disease [28]. Compared with late-onset PE, early-onset PE usually proceeds more aggressively, shows faster deterioration, and is commonly associated with fetal growth restriction, features that are attributed to early gestational dysfunction of the placenta [29]. By contrast, late-onset PE is more frequently associated with maternal phenotypic characteristics, such as obesity, diabetes, or metabolic syndrome, and is presumably less affected by placental dysfunction [29]. For these reasons, some authors have suggested classifying the two entities as placental PE and maternal PE, respectively [30]. Because there is considerable overlap between these two entities in both classification methods, maternal hemodynamic characteristics have been evaluated in PE-associated pregnancies relative to the presence or absence of (fetal) intrauterine growth restriction (IUGR) [31,32]. PE with IUGR, which predominantly occurs with the early-onset or placental type of PE, presents with signs of a vascular hypertonic state, whereas, in PE without IUGR (usually associated with the late-onset or maternal-type PE), hypertension is mainly related to high cardiac output [31,33]. This classification is consistent with the hemodynamic definition of high blood pressure, which, according to Ohm’s law, is “mean arterial pressure = cardiac output × peripheral resistance” [34]. Of all the PE classification systems, the latter method not only accommodates most fundamental physiological factors, but also offers a better opportunity for the more-targeted management of both subtypes [31,32,33]. To complete the picture, two different subtypes of late-onset PE have also been reported, based on the bimodal skewing of the birth weight percentiles and uterine artery Doppler ultrasound measurements [35]. The categorization of PE into three subtypes is not only consistent with the reported prevalence figures, which are higher for late-onset PE than for early-onset PE [36], but also with the three reported types of longitudinal changes in maternal cardiovascular functions throughout the course of the pregnancy, as will be explained below [37,38,39].

The upcoming paragraphs discuss one by one the role of the different functional processes and organ systems in the pathophysiology of preeclampsia, many of which present simultaneously. For readers’ comprehension, Table 2 presents the consecutive sequence in time of these events, with special reference to related figures. 

### 2.2. Placentation Process

#### 2.2.1. Spiral Artery Remodeling in Normal Pregnancy and Preeclampsia

The normal and pathological processes of uterine artery remodeling in the early stages of pregnancy have been described well and republished in basic textbooks on obstetrics as one of the fundamental concepts of gestational physiology. Normal placentation occurs in a two-step process, in which spiral arteries in the center of the placental bed become disrupted, invaded, and remodeled by trophoblasts between 8 and 14 weeks, followed by invasion of the junctional-zone myometrial vascular segments in the later stages [45]. The net result is a marked increase in the oxygen tension of the placental bed from 10 weeks onwards. Inadequate spiral artery remodeling was more frequent in placental bed biopsies after PE than after normal pregnancies, evident as the shallow dilatation of the arterial lumen, less-extensive invasion of the myometrial segments, and signs of hyperplastic arteriosclerosis and atherosclerosis [45]. The net result is a reduced oxygen supply to the placental bed, resulting in local oxidative stress, which gradually spreads throughout the global cardiovascular system as the pregnancy advances (see ‘Maternal–fetal communication’ below). The observation that the symptoms of PE resolve faster after early postpartum curettage of the uterine cavity suggests that the placenta acts as the primary driver of the cardiovascular dysfunctions in PE [46]. 

#### 2.2.2. Vascular Uterine Adaptation Involves More than Spiral Artery Remodeling 

There is now a growing body of evidence refuting abnormal placentation as a unifactorial etiology of PE [47]. A meta-analysis of blinded and unblinded placental histology studies showed villous and/or vascular placental lesions in 39–42% of PE compared with 10–19% in normal pregnancies, indicating that abnormal placentation is not a prerequisite nor pathognomonic for PE [48]. It has also been shown that uterine vascular changes occur before the first wave of trophoblast invasion, as documented in angiographic studies dating back to the 1960s, in which doubling of the uterine artery diameter was observed at 7 weeks of gestation. These data were later confirmed in experimental and histological studies [45,49]. More recent evidence has shown that veins and lymphatic vessels are invaded at an earlier stage than the spiral arteries, illustrating that the adaptation of the uterine circulation involves much more than just the arteries [50]. Consistent with theoretical models describing the increased flow velocities and oxygen tension in the intervillous space downstream from narrow spiral arteries [51], transabdominal near-infrared spectroscopic measurements of the oxygenation of placental tissue confirmed greater placental oxygenation during PE than during normal pregnancies [52]. This contradicts the concept of placental hypoxia as the primary trigger for the cascade of events that ultimately causes PE. In silico and in vitro perfusion models of the human placenta, with modifiable maternal artery perfusion pressures, have not only demonstrated increased intervillous hydrostatic pressure and associated flow changes when the perfusion rate is increased, but also morphological damage, such as vacuolization and the shedding of syncitiotrophoblast cells, similar to the lesions observed in PE [53,54]. Together with the physiological concept of a lower flow velocity and lower perfusion pressure downstream from increased resistance to the uterine arterial flow [55], these observations support the idea that abnormal spiral artery remodeling is a consequence, rather than a cause, of abnormal perfusion of the intervillous space (Figure 1). 

#### 2.2.3. Hemodynamics of the Intervillous Space

The concept of a mechanistic driving force of uterine artery (mal)adaptation is supported by a recent review of the functions of endothelial mechanosensitive Piezo 1 channels, which transduce shear stress into increased calcium influx and activation of the nitric oxide (NO) pathway, with subsequent uterine vascular dilatation [56]. Shear stress is an important driver of trophoblast-induced endothelial apoptosis during early placentation [57]. Mechanosensitivity in the earliest stages of placentation and throughout the course of pregnancy is consistent with a bidirectional model of maternal–fetal communication (see chapter maternal–fetal communication), and is consistent with the observations in maternal hemodynamic studies (see chapter maternal hemodynamics). This supports the view that placental hypoxia and oxidative stress are consequences of abnormal vascular adaptation. Placental hypoxia and reperfusion mechanisms trigger mitochondrial dysfunction and damage, and initiate the molecular mechanisms of oxidative stress, which is reportedly higher in PE than in uncomplicated pregnancies [58]. In turn, oxidative stress triggers intravascular inflammation and endothelial dysfunction via the release of factors such as tumor necrosis factor-α (TNFα), interleukin-6 (IL6), IL10, C-reactive protein, and other factors, such as Hemoglobin F [59]. Oxidative stress is also responsible for structural damage to DNA, proteins, and lipids long before the onset of clinical symptoms [58]. These effects are enhanced by epigenetic gene modifications, including altered histone modifications, DNA methylation, and the production of noncoding RNAs, such as microRNAs or circular RNAs [60]. Epigenetic gene modification not only occurs in placental tissue, but also in circulating leucocytes, shed extracellular trophoblast particles, and cell-free DNA and RNA. As will be discussed below, these play an important role in fetal–maternal communication and in triggering the systemic inflammatory response [60,61,62], in which they are assisted by the presence of trophoblast plugs in the uterine arteries. Before 10 weeks of gestation, the intravascular lumen of the spiral arteries is blocked by solid trophoblast plugs, containing numerous endovascular trophoblast cells and adhesive molecules, such as matrix metallopeptidase 1 (MMP1) [63]. These plugs are impenetrable to circulating blood cells but permeable to plasma, contributing to the nutrient supply to the embryo, which, at this stage, mainly depends on histiotropic cell to cell transfer of energy sources. The intra-arterial trophoblast plugs serve several important functions:-Maintenance of an oxygen gradient with a low embryonic oxygen tension, protecting the conceptus from toxic oxygen metabolites [63,64].-Reduction in perfusion pressure and flow velocity during the formation of the intervillous space, which are necessary for trophoblast cell invasion and the remodeling of the uterine vasculature (Figure 2C) [64,65].-Diversion of the arterial blood from the radial arteries to the uterine venules by arteriovenous anastomoses (Figure 2C). These vascular connections allow direct endothelial cell–cell communication between the uterine venular and arteriolar vessels during the formation of the intervillous space [55,65,66].-Initiation of the adaptation of the maternal systemic cardiovasculature by causing the drainage of endocrine and trophoblast signals into the maternal circulation. During blockage of the spiral arteries, trophoblast cells invade and dilate the maternal veins [64], a process that also contributes to maintaining the low pressure and flow conditions during the formation of the intervillous space [63].-In a similar way, a systemic maternal immune response and tolerance are generated by the trophoblast invasion of lymphatic vessels, allowing the drainage and presentation of immunoregulatory signals to immune cells at locations distant from the uterus [64].

When the trophoblast plugs start to degrade at around 10 weeks, both the uterine and systemic circulations and the immune system are adapted to accept and accommodate the further growth of the fetus and the placenta.

### 2.3. Immune System, Implantation, and Vascular Remodeling

The immune system plays an important role in the early stages of embryo implantation, balancing the pro- and anti-inflammatory responses to fetal paternal antigens with the establishment of controlled immune tolerance, while simultaneously maintaining an adequate defense against microorganisms [67]. The T-cell-mediated activation of complement and the proliferation of neutrophils are suppressed and the stroma is penetrated by tolerant uterine natural killer cells, while proinflammatory macrophages, mast cells, T-suppressor and T-regulatory cells are activated, in association with T-helper cell differentiation, the accumulation of T-regulatory cells in the uterine myometrium, and the control of B-cell functions (Figure 2B) [68]. The production of factors that mediate angiogenesis and vasculogenesis increases, such as vascular endothelial growth factor (VEGF), placental growth factor, Tumor Necrosis Factor (TNF) α, Interleukin (IL)1β, IL6, IL8, and Matrixmetalloproteases (MMP) [68,69,70]. MMPs play a role in the degradation of the extracellular matrix surrounding the uterine vasculature, as the first step in uterine artery remodeling, and are stimulated by coagulation regulators, such as urokinase plasminogen activator. The fibrinolytic functions of these proteins are, in turn, regulated by a plasminogen activator inhibitor [71]. Human chorionic gonadotropin is an important mediator of this complex process, during which numerous cytokines, ILs, and proangiogenic factors are produced, contributing to fetal and maternal angiogenesis, vasculogenesis, and uterine artery remodeling, as described above [68,70]. Compared with normal pregnancies, PE is associated with increased serum concentrations of complement factors C3a and sC5b-9 and chemokines CXCL10 and CXCL11, and reduced levels of IL4 [72]. Pentraxin-3 activity is also upregulated in PE. Pentraxin-3 is produced by macrophages, fibroblasts, dendritic cells, and endothelial cells to regulate NO synthesis, to activate complement, macrophages, and dendritic cells, and to promote trophoblast apoptosis [72,73]. Importantly, increased serum concentrations of C3a, CCXL10, and petraxin-3 are found in nulliparous but not multiparous women [72]. Furthermore, uterine and circulating T-regulator cells have a memory function for fetal–paternal antigens in multigravida women [67]. These observations illustrate the important contributions made by the maternal immune system to the mechanisms underlying the increased risk posed by primipaternity and the reduced risk of HIV-positive status in the development of PE [40]. Recent research strongly emphasizes that the quality of the earliest steps of implantation determines the quality of the ongoing pregnancy [67].

### 2.4. Fetal–Maternal Communication

#### 2.4.1. Pre- and Peri-Implantation Signaling

The crosstalk between the conceptus and the mother begins before implantation [74]. Animal studies have shown that during the conception–implantation interval, there is a four-fold increase in proteins in the blastocyst cavity, of which 20% are embryonic and 80% are maternal in origin. One of the earliest embryonic signals besides chorionic gonadotropin is proinflammatory IL1β (Figure 2A), which prepares the inflammatory environment required for the successful implantation of the embryo [75]. Insulin-like growth factor (IGF) produced by the mother promotes early embryonic development via IGF-binding proteins expressed by the preimplantation embryo [74]. Maternal products, such as haptoglobin and uteroglobin, are captured by the embryo and re-presented to the mother, where they play a role in modulating the maternal immune system [74].

#### 2.4.2. Endocrine, Biological and Chemical Signaling

Other endocrine mechanisms are involved in fetal–maternal communication and the maternal adaptation to pregnancy. Estrogens and progestogens not only stimulate local placental and uterine angiogenesis, but also contribute to maternal systemic vasodilatation by increasing prostaglandin I2 (PGI2) and NO production [76,77]. The net results are increased uterine blood flow and reduced blood pressure. Since the introduction of highly specific laboratory assays, there is increasing evidence that PE is associated with altered serum hormone concentrations, with an increased androgen/estrogen balance, which contributes to unbalanced vasoactivity [69,76,77]. Estrogens, and to a lesser extent progestogens, exert their vascular effects by increasing the intracellular calcium concentration, and its swift intercellular spread through increased gap junction communication [69]. Intracellular calcium bursts are required for the catalysis of NO, PGI2, and endothelium-derived hyperpolarizing factor (EDHF). NO and PGI2 diffuse from the endometrium to the vascular smooth muscle layer, where they promote the production of cyclic nucleotides cGMP and cAMP, which both act as strong inhibitors of vascular tone. These cyclic nucleotides are active at both local and distant locations, increasing intracellular calcium bursts and gap junctions, before or after their release into the circulation. In normal pregnancies, these self-reinforcing activities build up gradually towards sustained vasodilatation throughout the circulation. As is discussed above, in PE, the cellular products of inflammation and circulating factors, such as placental growth factor, soluble fms-like tyrosine kinase 1 (sFLT1), TNFα, ILs, and many others, cause major interference. The net effect is a state of vasoconstriction arising from reduced intracellular calcium signaling, the closure of gap junctions, and the disrupted integrity of the endothelium monolayer, presenting clinically as hypertension, edema, proteinuria, and other organ dysfunctions [69].

#### 2.4.3. Syncytiotrophoblast Extracellular Vesicles

As mentioned above, an important fetal–maternal communication system involves the intravascular shedding of placental particles, varying in size and shape from multinucleated syncytial aggregates to subcellular nanovesicles, originating from apoptosis in normal pregnancies, but also from necrosis in PE [78]. This phenomenon is also associated with increased serum levels of total cell-free DNA [79]. Both in vitro and in vivo animal studies have shown that these particles are cleared from the circulation via phagocytosis by macrophages and endothelial cells at locations distant from the uterus, where they induce a reactive endothelium cell response. In uncomplicated pregnancies, endothelial cells become progressively less sensitive to vasoconstrictive mediators. However, in PE, they show signs of activation, including increased surface expression of monocyte adhesion receptors, such as E-selectin, and the secretion of proinflammatory IL6 and transforming growth factor β (TGF-β). This activation process spreads rapidly via paracrine and endocrine pathways to other nearby or distant endothelial cells [78]. As discussed above, these particles may act via intravesicular microRNAs and/or circular RNAs that, after phagocytosis by endothelial and immune cells, induce sterile inflammation and alter the production of mediators of angiogenesis and vasoactivity, such as sFLT1, VEGF, and pregnancy-associated placental protein A (PAPP-A) [61,62,80]. Many other vasoactive and immunological mediators have been studied in the maternal serum for the diagnosis or prediction of PE, and are summarized in Table 3. It is generally accepted that the placenta is the primary source of these factors and is therefore the key driver of the global functioning of the maternal circulation [81]. However, it should be emphasized that abnormal serum concentrations of many of these factors have also been documented in nonpregnant individuals with (preclinical) chronic cardiovascular and/or renal disease (Table 3). This indicates that, apart from some gestation-specific products, it is still unclear whether the origin of the serum vasoactive and/or immunomodulatory substances associated with PE is placental, maternal, or both.

### 2.5. Maternal Hemodynamics

#### 2.5.1. Peri-Implantation Hemodynamics

The very first change that occurs in the maternal circulation is a generalized reduction in vascular tone with subsequent vasodilatation, triggering volume retention mechanisms to expand the circulating volume [106]. This is associated with an increase in cardiac output in the very first weeks after implantation [39], coincident with an increase in the volume of the intrathoracic fluid [107]. It should be emphasized that, in these very early stages, uterine spiral artery remodeling is still incomplete and the trophoblast plugs are still in place. In pregnancies complicated with hypertension, particularly those associated with impaired fetal growth, the early-gestational increase in cardiac output is less profound because the increase in the stroke volume is shallow [39,108]. This reduces the expansion of the plasma volume in PE compared with that in normal pregnancies [109]. From this, it can be concluded that the cardiovascular conditions that predispose a woman to a normal outcome or to gestational hypertensive disorders, with or without fetal growth restriction, are already present in the first weeks after implantation. Furthermore, abnormal cardiac output and/or peripheral arterial resistance have been documented before conception in some women who subsequently developed PE or fetal growth restriction [41]. This supports the view that peri-implantation maternal cardiovascular dysfunction is the cause, rather than a consequence, of abnormal placentation (Figure 1) [110].

#### 2.5.2. Body Water Volume Expansion and (Subclinical) Cardiovascular Dysfunctions

Irrespective of the gestational outcome, the body water volume increases in all women during pregnancy [111]. Body water includes both circulating and noncirculating volumes, among which are the plasma volume, intra- and extracellular water, and interstitial fluid (interstitial fluid = extracellular water – plasma volume) [112]. The expansion of the intravascular volume is an important stressor of the maternal cardiovascular system, as illustrated by the large proportion of women who show cardiac signs of volume overload during uncomplicated third-trimester pregnancies [113]. This observation is particularly evident in obese women, whose body water volume and cardiac output are greater than those of normal-weight women [42]. It has also been shown that when this volume expansion is superimposed upon first-trimester (subclinical) dysfunctions of one or more elements of the maternal cardiovascular circuit, a gradual deterioration in the global circulatory function may occur. This process is unique for each type of GHD, eventually presenting as either GH, early-, or late-onset PE (Figure 3) [114]. It is important to emphasize that the type of hypertensive disorder that occurs is related to the type of early-gestational dysfunction, implying that the biophysical parameters determined by noninvasive assessment of the maternal hemodynamics are useful in screening for gestational hypertensive disorders [115]. The mechanisms of this gradual circulatory deterioration are very similar to those involved in cardiorenal syndrome, where interorgan communication occurs via endocrine, metabolic, endothelial, immunological, and autonomic nervous mechanisms [44]. Cardiorenal communication mechanisms help explain the different clinical phenotypes of gestational hypertensive disorders: pre- and/or periconceptional cardiovascular dysfunction predisposes the pregnant woman to early-onset PE, as in chronic cardiorenal syndrome type II. In some late-onset PE, however, endothelial dysfunction may develop as a result of volume overload, as in acute cardiorenal syndrome type I [44]. An important point of interest here is that the involvement of the maternal venous compartment in global circulatory dysfunction has been observed in PE but not in GH, suggesting that inadequate volume regulation and venous hemodynamic functions may be a much more pathophysiologically important aspect of PE than is presently believed [116,117].

#### 2.5.3. Venous Hemodynamics 

The venous compartment serves three important physiological functions: (1) the regulation of the cardiac output, in close cooperation with the heart; (2) the storage of a noncirculating reserve blood volume, mainly in the splanchnic and liver beds; and (3) the control of capillary function by maintaining the high-volume/low-pressure venous return. The relevance of these functions during pregnancy is demonstrated by the correlations observed between maternal cardiac output, the Doppler parameters of the hepatic venous hemodynamics, and the neonatal birth weight [118]. Consistent with this, a constitutionally low body water volume and cardiac output, in association with activated venous return mechanisms, were recorded throughout gestation in women who gave birth to neonates that were small for gestational age [43]. As explained above, low and high cardiac outputs distinguish the two phenotypes of PE, one presenting with fetal growth restriction and the other with a normal-to-high neonatal birth weight, respectively [31,33]. Abnormal maternal venous Doppler parameters are more pronounced and show a longer preclinical latency phase in early-onset PE than in late-onset PE [119,120]. These observations suggest that a more activated state of venous hemodynamics supports the cardiac output in low-output PE more than in high-output PE, a characteristic not observed in GH [116,117]. This suggestion is corroborated by the reported association between PE and congenital maternal heart disease with right heart dysfunction [121], and with the predisposition to later-life diastolic dysfunction in these women, eventually evolving to heart failure with a preserved ejection fraction [122]. Therefore, these observations support the view that venous congestion is a key phenomenon explaining the clinical picture of PE as “a GH syndrome with signs of dysfunction in one or more organ” [2,3].

#### 2.5.4. Venous Congestion

Venous congestion is a microcirculatory dysfunction with unbalanced capillary leakage/resorption of the intravascular contents during conditions in which the arterial blood supply exceeds the venous outflow. This latter phenomenon can result from increased intravenous pressure or venous hypertension, from a reduced forward venous flow velocity evolving to stasis of the blood, and from external venous compression (Figure 4) [123]. Smooth muscle constriction in the venous vascular wall increases the vascular tone and causes intravenous hypertension. The Doppler characteristics observed during early-onset PE are consistent with increased venous vascular tone [119], a condition predisposing a woman to poor venous return in the long run [114]. This is associated with poor expansion of the plasma volume [109], a shallow gestation-induced increase in the cardiac output [39,111], a profound increase in peripheral vascular resistance [39] and uterine artery impedance [33,114]. This condition leads to poor fetal growth and ultimately to early failure of one or more maternal organs and/or fetal distress, for which early delivery of the baby is usually the only solution.

Inadequate venous return and/or organ drainage can be a consequence of poor cardiac diastolic function, with or without intravascular volume overload or (sub)obstructed venous outflow. In pregnancy, these conditions often present with maternal obesity, diabetes, and metabolic syndrome, and evolve to maternal or late-onset PE with an appropriate or large-for-gestational-age neonatal birth weight [33,114]. Longitudinal observations throughout pregnancy show an early-gestational cardiovascular state of high flow/low resistance, which is converted to a low flow/high resistance state in advanced pregnancy (Figure 5) [38]. This conversion probably results from endothelial activation caused by intravascular overload, similar to that in cardiorenal syndrome type I, as discussed above, with or without enhancement by the placental oxidative stress that develops during the course of the pregnancy [124].

#### 2.5.5. The Role of Intra-Abdominal Pressure

Increased intra-abdominal pressure during pregnancy is a well-documented but poorly recognized physiological phenomenon [126,127], with important external compression effects on the venous return in maternal intra-abdominal veins. In nonpregnant women, intra-abdominal hypertension reduces the venous return and cardiac output, with a reflex-induced increase in the arteriolar tone and reduced organ perfusion [128]. This condition gradually evolves into a state of intra-abdominal compartment syndrome with multiorgan failure and is only resolved after the release of the intraabdominal pressure [128]. Abdominal compartment syndrome is often associated with intravascular overload within the context of an originally normal but gradually deteriorating cardiovascular function [129]. This situation is also present in the term pregnancies of obese women or those with a large uterine volume resulting from multiplets or polyhydramnios, which are all known risk factors for PE, as discussed above. Longitudinal observations of persistently high cardiac output throughout all stages of pregnancy, including the latent and clinical phase of late-onset PE, were reported by Easterling in a cohort of mainly obese women (Figure 5) [37]. These observations were consistent with a state of gestation-induced intra-abdominal hypertension and subsequent organ dysfunction resulting from venous congestion, a condition that is also observed during laparoscopy-induced pneumoperitoneum [130].

## 3. Integrated Pathophysiology of PE

From the information summarized in this paper, it is clear that pregnancy—the process from the implantation of the conceptus to the delivery of the baby—is a multistage stepwise accumulation of key events that all serve one primary goal: the survival of the heir. These key events can be condensed to: (1) control the maternal immune system; (2) bidirectional conceptus–mother communication; and (3) adaptation of the maternal systemic functions. A schematic summary of these pathophysiologic keystones is enlisted in Table 2, and a pictorial summary is presented in Figure 6.

### 3.1. Immunomodulation

Immunology-mediating factors are already released by the conceptus in the preimplantation stage, to promote the intrauterine inflammatory state that facilitates successful implantation [74,75]. This process is continued after implantation and supplemented with a second immunomodulatory effect: the maternal tolerance of the paternal antigens carried by the implanting embryo [75]. Inadequate signaling by the conceptus may result in an imbalanced inflammatory response and the subsequent maladaptation of the intrauterine vasculature [74]. The role of the immunomodulatory conceptus is illustrated by the fact that primipaternity is a risk factor for the development of PE, and in the observation that males can father preeclamptic pregnancies in multiple partners [9]. The role of the mother is illustrated by the fact that the risk of PE is increased in those with autoimmune disorders, such as antiphospholipid syndrome [13], with potential contributions by genetic and racial factors [7,8,10,11,12].

### 3.2. Maternal-Fetal Crosstalk

The first stages of distant fetal–maternal communication occur via trophoblast invasion of the intrauterine veins and lymphatic vessels [50], during the stage when the spiral arteries are blocked with trophoblast plugs [63,64,65]. Before the activation of the intervillous arterial supply, biochemical and epigenetic signals are released into the maternal system, preparing the female organ systems to accept and accommodate the growth of the fetus [60,61,62]. Errors in this communication process can be bidirectional because the fetal signals may be inadequate and/or the maternal response may be unbalanced. The recurrence of PE in pregnancies fathered by different partners illustrates a maternal failure in this communication process. Inadequate venous or lymphatic preparation of spiral artery remodeling can partly explain the simultaneous occurrence of the shallow dilatation of the spiral arteries and the increased impedance of the uterine artery as two important contributors to the best possible flow conditions for maternal–fetal exchange in the intervillous space [55].

### 3.3. Maternal Hemodynamics and Venous Congestion

The key role of maternal cardiovascular adaptation in the normal course of pregnancy has been known for a long time. This adaptation not only involves the maternal heart and arteries, but also the expansion of the intravascular volume and the venous compartment. Inadequate venous hemodynamics, hampering the drainage of blood from the organs, predisposes the woman to venous congestion, presenting as edema and proteinuria, which among other factors, are key features of PE. Venous congestion also partly explains the semi-identical phenotypes of PE, despite different pathophysiological mechanisms in the latent phase [114], as well as the different types of longitudinal gestational evolution of the maternal cardiovascular function, illustrated in Figure 5 [37,38,39]. Preexisting diseases, such as the essential hypertension or renal dysfunction, are well-known predisposing factors for the development of PE [13]. However, it has only recently become clear that subclinical cardiovascular dysfunction may also be present in apparently healthy individuals with normal blood pressure and heart rate, predisposing them to the development of PE [33]. Screening for and the diagnosis of these subclinical dysfunctions before conception will be important challenges for prenatal healthcare workers and researchers in the near future, with the aim of reducing and/or even preventing PE, which is one of the most important medical threats for young women and their offspring.

## 4. Conclusions

This review integrates information from many other reviews of the molecular, cellular, and systemic pathophysiological mechanisms involved in the clinical gestational syndrome PE. In particular, the relevance of immunological modulation, maternal–fetal communication, and maternal cardiovascular adaptation as independent and interdependent processes is highlighted, with equally important contributions from both the conceptus and the mother. Incorrect signaling or responses on either side can explain the conditions that predispose a pregnancy to PE and the different pathways to the development of a common clinical phenotypic syndrome. These present as new-onset hypertension combined with the symptoms of failure in one or more organs. Together, the information summarized in this review demonstrates that PE is not a unicausal disorder but a multifaceted, multifactorial syndrome—fundamentally not unique to pregnancy—and that no aspect must be overlooked in the research, management, screening, and prevention of this ever-devastating threat to maternal and neonatal health.

## Figures and Tables

**Figure 1 jcm-09-02245-f001:**
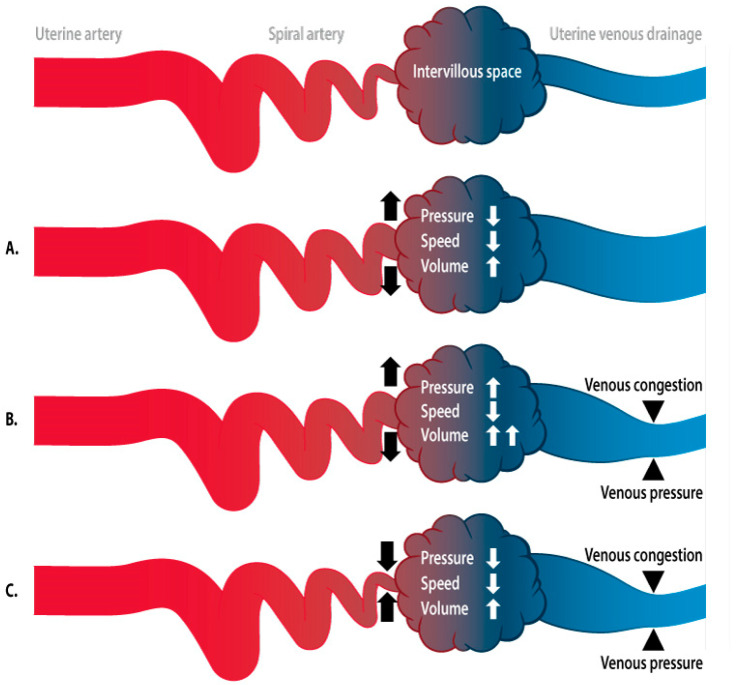
Schematic presentation of the hemodynamics of the intervillous space. As in any capillary network, the functioning of the intervillous space is determined by a balanced intra- and extravascular exchange under flow conditions that are tightly regulated by arteriolar and venular activity in accordance with osmotic forces of the perivascular tissue. (**A**): Hemodynamic effects in the intervillous space caused by spiral artery dilatation. In this condition, the flow volume increases, but flow velocity and pressure decrease. This condition allows optimal maternal–fetal exchange. (**B**): Hemodynamic effects in the intervillous space when spiral artery dilatation occurs after inadequate adaptation of the venous outflow. There is stasis of blood that cannot be drained efficiently and causes congestion, responsible for increased pressure and reduced flow velocity in the intervillous space. (**C**): Impact of the incomplete dilatation of the spiral arteries on the conditions presented in Figure 1B. The reduced spiral artery blood supply reduces the degree of congestion and pressure in the intervillous space, at the cost of a reduced flow volume. This mechanism might improve the maternal–fetal exchange when there are unbalanced pressure/flow conditions in the intervillous space as in Figure 1B.

**Figure 2 jcm-09-02245-f002:**
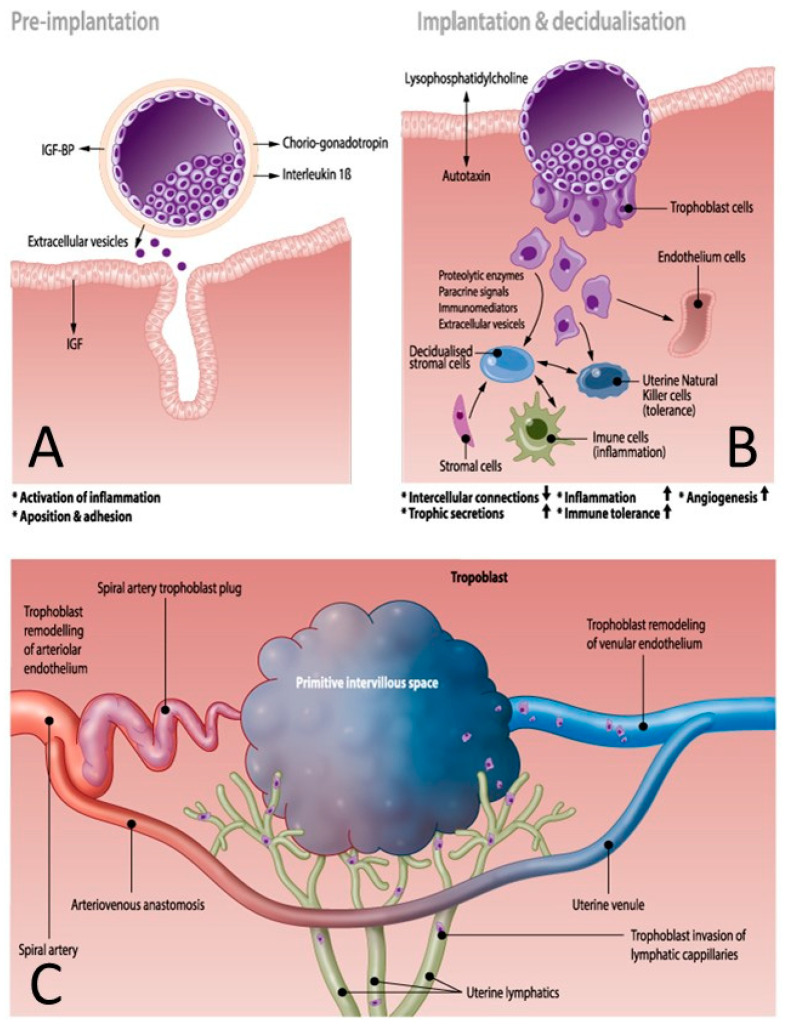
Early maternal and uterine vascular adaptations before the initiation of spiral artery flow. (**A**): Before implantation, the conceptus sheds products such as choriogonadotropin, interleukins, insulin-like growth factor (IGF) and extracellular vesicles. Proinflammatory interleukin 1β activates an endometrial inflammatory response, promoting apposition and adhesion at the implantation site with assistance of maternal insulin-like growth factor (IGF). (**B**): Fetal–maternal communication during implantation is illustrated by the release of embryonic lysophosphatidylcholine, which is metabolized by maternal autotaxin, promoting the invasion of the conceptus and the initiation of decidualization. This process involves loosening the intercellular connections and the release of trophic secretions, together with signals promoting inflammation, immune tolerance, and angiogenesis. Trophoblast cells modulate the functions of stromal cells, immune cells, natural killer cells, and endothelial cells by releasing proteolytic enzymes, paracrine and immunomodulatory agents, and extracellular vesicles. (**C**): Early stages of maternal vascular adaptation during the formation of the intervillous space. Spiral arteries are obstructed with trophoblast plugs and arterial blood is directed to the venous system via arteriovenous anastomoses. Simultaneously, the venular and lymphatic microcirculations are penetrated and modulated by trophoblast cells, promoting the release of gestational products into the maternal circulation and lymphatic system, weeks before the spiral artery flow actually starts. During this stage, the fetal environment is very hypoxic and growth of the conceptus is supported by maternal plasma products penetrating the trophoblast plugs, together with decidual histiotrophy. The three panels of this figure show that the abnormal adaptation of the uterine vasculature can result from an unbalanced inflammatory reaction, caused by inadequate signaling by the conceptus or a suboptimal maternal response. Abnormal remodeling of the venous and lymphatic vessels is responsible for the inefficient systemic distribution of adaptation signals and the inappropriate formation of the early intervillous space at the stage in which the spiral artery flow has not yet started.

**Figure 3 jcm-09-02245-f003:**
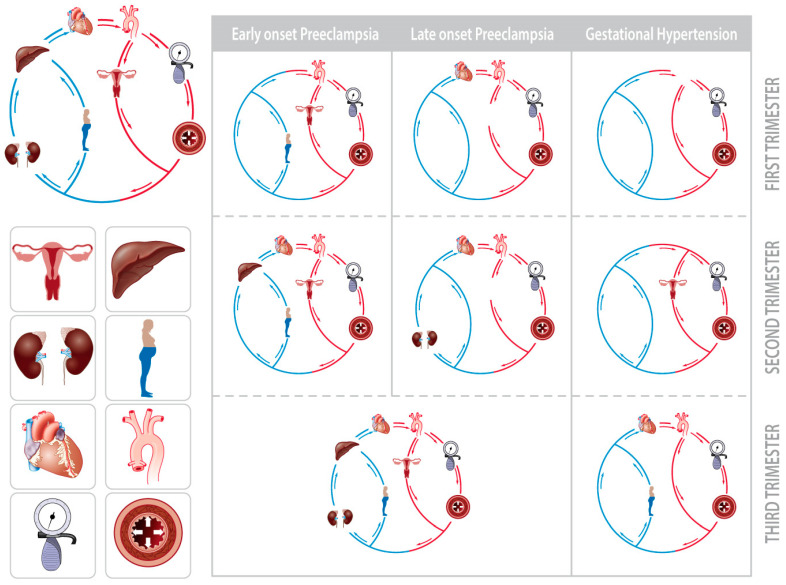
Schematic presentation of the gradual deterioration of circulatory dysfunction from the first to third trimester in early-onset preeclampsia, late-onset preeclampsia, and gestational hypertension (adapted from [114]). The circulation is presented as a closed circuit, with interdependently functioning components. The depicted icons in the circuit represent the sites/organs where abnormal hemodynamic characteristics are measured using noninvasive impedance cardiography, bioimpedence spectrum analysis, and combined ECG/Doppler ultrasonography [114]. Early gestational dysfunctions in the first trimester gradually increase throughout the course of the pregnancy, and this co-exists with the expansion of body water volumes [111]. The final end stage, defined as early-onset preeclampsia, late-onset preeclampsia, or gestational hypertension, depends on the type of hemodynamic dysfunction already present in the first trimester. Note that abnormal Doppler characteristics of the hepatic and renal interlobar veins are observed in preeclampsia but not in gestational hypertension.

**Figure 4 jcm-09-02245-f004:**
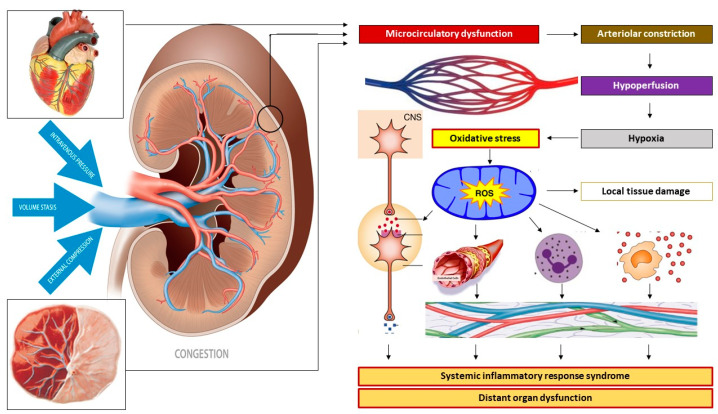
Mechanisms of venous congestion at the level of the kidney. Venous congestion can result from venous vascular hypertonia with subsequent venous hypertension, from volume stasis arising from (sub-)obstructed outflow, and/or from external compression. At the level of the kidneys, this congestion induces a retrograde dysfunction of the peritubular and glomerular capillaries. Reflex arteriolar constriction and activation of the renin–angiotensin–aldosterone system prevents further microcirculatory deterioration at the cost of reduced organ perfusion and subsequent parenchymal hypoxia. An oxidative stress response induces immunological, neurological, and metabolic protection mechanisms by activating the autonomic nervous system and the intravascular distribution of cytokines, endocrines, and vasoactive mediators, causing endothelial activation and a generalized state of inflammation. In non-pregnant individuals, this sequence of events unfolds in the kidneys as part of the pathophysiology of renocardial syndrome but can also occur in other internal organs as in cardiohepatic or hepatorenal syndromes. In pregnancy, this mechanism may unfold in the uterus and placenta as well as in internal organs. In this picture, the kidney may serve as a proxy for other maternal organs, such as uterus-placenta, heart, etc.

**Figure 5 jcm-09-02245-f005:**
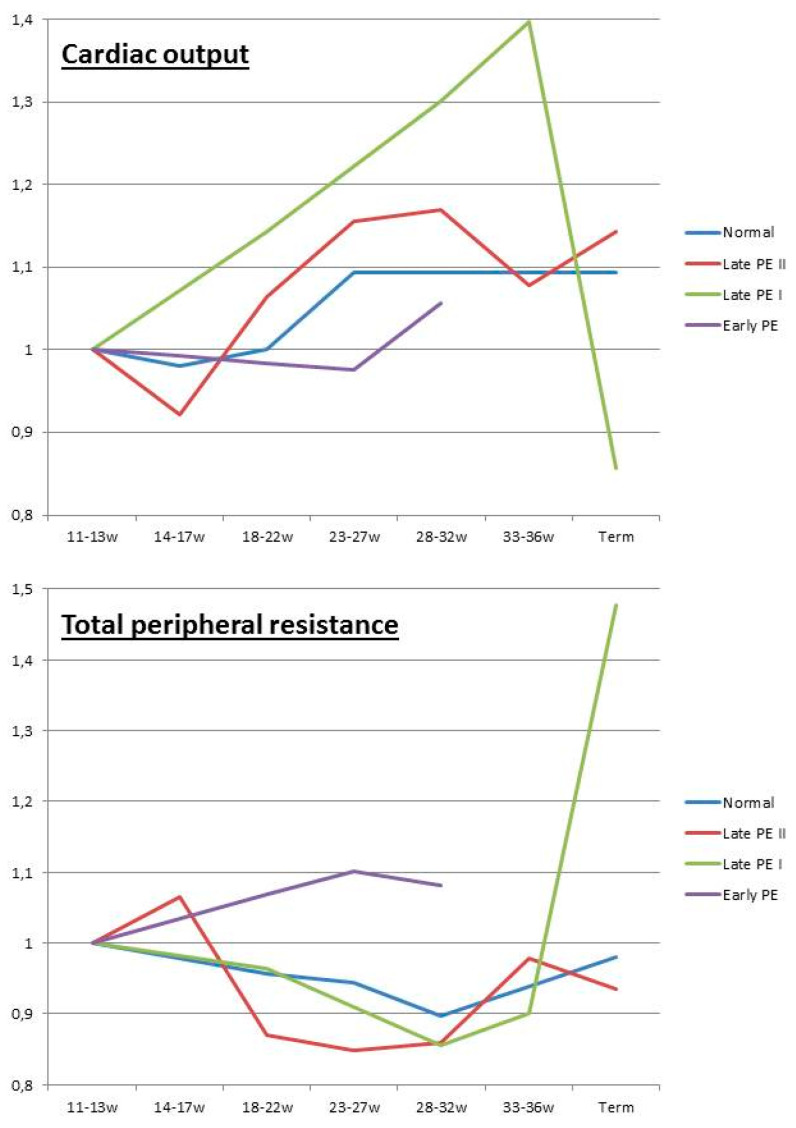
Longitudinal changes in cardiac output and peripheral resistance, expressed as a product of 12-week measurements, reported in normal pregnancies [125], early-onset preeclampsia [114], late-onset preeclampsia type I (cross over) [38], and late-onset preeclampsia type II (high output) [37]. In normal pregnancies, the cardiac output increases to a plateau from the second trimester until term, whereas the total peripheral resistance reaches a nadir in the early third trimester to increase again until term [125]. Early-onset preeclampsia presents with high total peripheral resistance from the first trimester onward, combined with a poor increase in cardiac output [114]. Late-onset preeclampsia type I presents with a normal change in peripheral resistance, but a more pronounced increase in cardiac output until the third trimester, when a fast crossover occurs from a high output/low resistance circulation to a low output/high resistance circulation [38]. Late-onset preeclampsia type II shows high cardiac output and low peripheral resistance throughout all stages of pregnancy in a population of mainly obese women [37].

**Figure 6 jcm-09-02245-f006:**
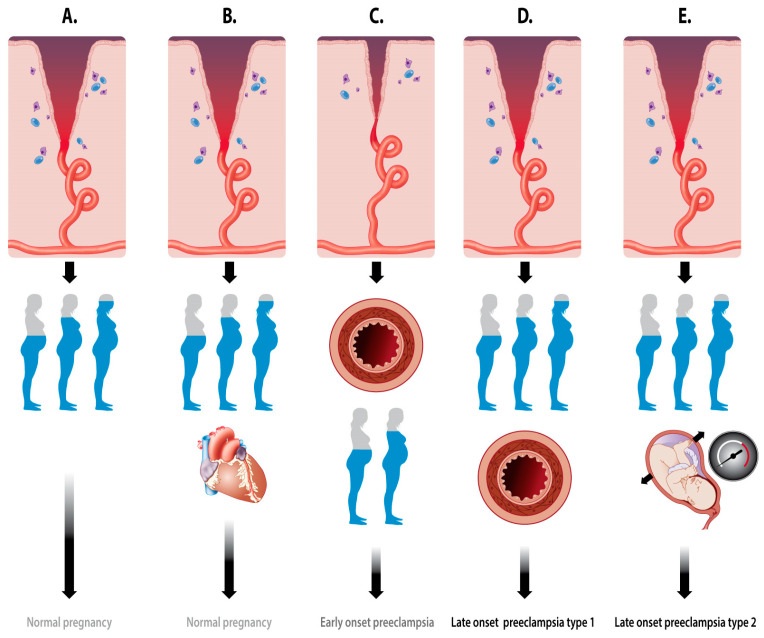
Schematic representation of the evolution from placentation and maternal vascular adaptation to the clinical end stage of (**A**) normal pregnancy, (**B**) normal pregnancy with cardiac signs of volume overload, (**C**) early-onset preeclampsia, (**D**) crossover late-onset preeclampsia (type I), and (**E**) high-output late-onset preeclampsia (type II). A: The normal remodeling of the spiral arteries is followed by the normal expansion of the body water volume in an uncomplicated pregnancy. B: The same situation as in panel A, but the woman presents at term with echocardiographic signs of volume overload [113]. C: Shallow spiral artery dilatation, presenting already in the first trimester with signs of endothelial activation and vascular hypertonia, followed by the suboptimal expansion of the plasma volume, and resulting in early-onset preeclampsia and poor fetal growth [39]. D: The normal remodeling of the spiral arteries is followed by the normal expansion of the body water volume with subsequent secondary endothelial activation, resulting in crossover from a high volume/low resistance to a low volume/high resistance circulation, eventually presenting as late-onset preeclampsia type I [38]. E: The normal remodeling of the spiral arteries is followed by the normal expansion of the body water volume in women with high intra-abdominal pressure, resulting in high-output late-onset preeclampsia type II [37].

**Table 1 jcm-09-02245-t001:** Criteria for the diagnosis of preeclampsia, as defined by the American College of Obstetrics and Gynecology (ACOG) and by the International Society for Studies of Hypertension in Pregnancy (ISSHP), with further specifications used by ACOG, Hypertension Canada (HT Canada), European Society of Cardiology (ESC), Society of Obstetricians and Gynaecologists of Canada (SOGC), ISSHP, Society of Obstetric Medicine of Australia and New Zealand (SOMANZ), and the Royal College of Obstetricians and Gynaecologist (RCOG [2,3,4]).

Hypertension (HT):	Systolic blood pressure ≥ 140 mm Hg and/or					
	diastolic blood pressure ≥ 90 mm Hg					
Chronic hypertension (CH):	hypertension before conception or					
	diagnosed ≤ 20 weeks of gestation					
GH (GH):	de novo hypertension after 20 weeks of gestation, without any signs of organ dysfunction					
PE (PE):		de novo HT ≥ 20 weeks of gestation, with ≥1 of following signs:				
ACOG 2019		ISSHP 2018				
HT 2 x ≥ 4 h apart	HT 2 x over few hours					
Proteinuria ≥ 300 mg/24 h						
Protein/creatinine ≥ 30 mg/mmol						
Dipstick ≥ 1+		Dipstick ≥ 2+ (> 1g/L)				
Liver transaminases > 2 × normal						
Platelets < 100 E^9^/L		Platelets < 150 E^9^/L				
Creatinine > 100 µmol/L		Creatinine ≥ 90 µmol/L				
Subjective signs						
		Uteroplacental dysfunction				
ACOG	HT Canada	ESC	SOGC	ISSHP	SOMANZ	RCOG
2019	2018	2018	2014	2018	2014	2011
CH	CH	CH	CH ± Sympt	CH 1°/2°	CH 1°/2°	CH
GH	GH	GH	GH ± Sympt	GH	GH	GH
PE/Ecl	PE/Ecl/HELLP	PE	PE	PE, CH→PE	PE/Ecl	PE
CH→PE		CH→PE	Transient HT	Transient HT	CH→PE	
		UHT	Wh Coat HT	Wh Coat HT	Wh Coat	
			Masked HT	Masked HT		
						HELLP
						Ecl
			Mild			Mild
		140-159/				140-149/
		90-109				90-99
						Moderate
						150-159/
						100-109
Severe	Severe	Severe	Severe	Severe	Severe	Severe
≥160/110	≥160/110	≥160/110	≥160/110	≥160/110	≥160/110	≥160/100
		Emergent	≥15 min	≥15 min	over few h	
		≥170/110				

Legend to Table 1: HT: Hypertension; CH: Chronic hypertension; GH: Gestational hypertension; PE: Preeclampsia; Ecl: Eclampsia; CH→PE: Preeclampsia superimposed on chronic hypertension; HELLP: Hemolysis, elevated liver enzymes, low platelets; UHT: Undefined hypertension; CH ± Sympt: Chronic hypertension with or without symptoms; GH ± Sympt: Gestational hypertension with or without symptoms; Transient HT: Transient hypertension; Wh Coat HT: White-coat hypertension; Masked HT: Masked hypertension; CH 1°/2°: Primary or secondary chronic hypertension.

**Table 2 jcm-09-02245-t002:** Consecutive sequence of pathophysiologic events associated with preeclampsia.

	Pathophysiologic Process	Figure	Predisposing Factors/Mechanisms	Clinics [references]
Inflammation	Preimplantation fetal signaling	2A	Genetic predisposition	Paternal preeclampsia [8,9]
	Maternal response	2B	Preconceptional paternal antigen immune memory	Primipaternity [6,9]
Local adaptation	Local maternal immune tolerance	2B	Maternal immunologic precondition	Auto-immune disease [13,14]
	Local inflammation	2B	Maternal immunologic precondition	Auto-immune disease [13,14], immune suppression [40]
	Local vascular adaptation	2B	Maternal cardiovascular/endothelial precondition	CV disease [13,14], endoth dysfunction [18]
Systemic adaptation	Systemic maternal immune tolerance	2C	Maternal immunologic precondition	Auto-immune disease [13,14]
	Systemic cardiovascular adaptation	2C	Maternal cardiovascular/endothelial precondition	CV disease [13,14], endoth dysfunction [18]
Early oxidative stress	Local tissue damage & inflammation	4	Maternal (immunologic) precondition	Auto-immune disease [13,14]
	Intercellular mediators, chemokines	4	Genetic/immunologic precondition	Genetic diseases [7.8.10.11], auto-immune diseases [13,14]
IVS hemodynamics	Veno-lymphatic remodelling & Arterial plugging	2C	Venous/ body volume dysfunction	(Subclinical) CV disease [41], body constitution [42,43]
		2C	Maternal immunologic dysfunction	Auto-immune disease [13,14]
	Spiral artery remodelling	1	Maternal arterial/endothelial dysfunction	CV disease [13,14], endoth dysfunction [18]
		1	Arterial response to poor venous adaptation	CV disease [13,14], endoth dysfunction [18]
Systemic Hemodynamics	Body water volume	3	Preconceptional low/high body water volume	Maternal body constitution [83.93], renal disease [13.14]
	Body water expansion upon early CV dysfunction	3	Extravasation/poor intravascular expansion	CV disease [13,14], renal disease [13.14], endoth dysfunction [18]
	Volume induced endoth dysfunction	3	Maternal constitutional/CV/endoth precondition	CV disease [13,14], renal disease [13.14], endoth dysfunction [18]
	Raised venous tone/venous hypertension	4	Maternal precondition	Aut NS [44], endoth dysfunction [18]
	External venous compression	4	Preconceptional raised intra-abdominal pressure	Maternal body constitution [83.93], renal disease [13.14]
		4	Increased gestational rise of intra-abdominal pressure	Large uterine volume [13,14]
Late oxidative stress	Intercellular mediators, chemokines	4	Genetic/immunologic precondition	Genetic [7.8.10.11], maternal disease [13.14], endoth [18], Aut NS [44]
	Systemic inflammation	4	Genetic/immunologic precondition	Genetic [7.8.10.11], maternal disease [13.14], endoth [18], Aut NS [44]
		4	Developing during pregnancy	Genetic [7.8.10.11], maternal disease [13.14], endoth [18], Aut NS [44]

CV: cardiovascular; Endoth: endothelium; Aut NS: autonomic nervous system; IVS: intervillous space.

**Table 3 jcm-09-02245-t003:** Serum markers of preeclampsia and/or cardiovascular disease.

	Physiologic Effects	PE	Non Pregnant	Type CVD	Ref
CRP	Immunomodulation	↑			[82]
			↑	CHD, HF	[83]
VEGF	Pro-angiogenic	↓			[84]
	Pro-vasculogenic		Polymorphisms	CHD	[85]
sFlt-1	Anti-angiogenic	↑			[84]
			↑	CHD, HF	[86]
sEng	Anti-angiogenic	↑			[84]
			↑	CHD	[87]
Gal-3BP	Immunomodulation	↑			[88]
			↑	CHD	[89]
Activin A	Immunomodulation	↑			[90]
	Apoptosis		↑	HT, CHD	[91]
Leptin	Immunomodulation	↑			[92]
	angiogenetic		↑	CHD	[93]
sE-selectin	Immunomodulation	↑			[94]
			↑	HT	[95]
ADAM 12	angiogenetic	↑			[96]
	immunomodulation		↑	HT	[97]
ADMA	vasodilatation	↑			[98]
			↑	CHD, HF, HT	[99]
PLGF	Pro-angiogenic	↓			[84]
			↑	CHD, HF	[86]
PAPP-A	Proteolysis IGF-BP	↓ → ↑			[94,100]
			↑	CHD	[101]
ADM	Pro-angiogenic	↓			[102]
	Vasodilatation		↑	AMI	[103]
PP13	Immunomodulation	↓			[104]
Inhibin A	Undetermined in pregnancy	↑			[90]
E3	Undetermined in pregnancy	↓			[105]
AFP	Undetermined in pregnancy	↑			[100]
HbF	A1M Immunomodulation	↑			[59]

Legend to Table 3: PE: Preeclampsia; CVD: Cardiovascular disease; ↑: High serum concentration; ↓: Low serum concentration; ↓ → ↑: Serum concentration changing from low to high; CHD: Coronary heart disease; HT: Hypertension; HF: Heart failure; AMI: Acute myocardial infarction; CRP: C-reactive protein; VEGF: Vascular endothelial growth factor; sFLT1:Soluble fms-like tyrosine kinase 1; sENG: Soluble endoglin; GAL-3BP: Galectin 3 binding protein; sE-selectin: Soluble E-selectin; ADAM12:A disintegrin and metalloproteinase 12; ADMA: Asymmetric dimethylarginine; PLGF: Placental growth factor; PAPP-A: Pregnancy-associated placental protein A; ADM: Adrenomedullin; PP13: Placental protein 13; E3: Estriol; AFP: Alpha fetoprotein; HbF: Fetal hemoglobin; A1M: Alfa-1-microglobulin.

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
