# Peer review of "Preeclampsia Is a Syndrome with a Cascade of Pathophysiologic Events"

_jcm, 2020, doi:10.3390/jcm9072245_

Round 1

Reviewer 1 Report

 Preeclampsia : a cascade of pathophysiologic events.

By:

 Wilfried Gyselaers

Originality:

Part of the theories is novel and important.

Quality:

The overall quality is high, and the review is comprehensive. There are however some points that need to be addressed;

  1. Among the less-frequently reported factors, the author list is air pollution. There is actually a growing bulk of reports confirming this association, also on a molecular level. This risk factor should be listed under environmental factors and perhaps a comment made in relation to the increasing incidence.
  2. The two-stage model by Redman is well established. There is an updated version by Redman/Staff that would be more appropriate and necessary to include. This model suggests an explanation for late-onset PE. The author could perhaps join the venous overflow hypothesis with the “pot bound” placenta-if the venous return from the placenta is impaired, this would affect placental function. The problem ought to become more significant as the placenta grows, hence this mechanism is more important in late-onset PE.
  3. Placental hypoxia has been described by Sothil et al. but uneven perfusion of the placenta is likely more important for the oxidative stress. Hypoxia as a factor is controversial, this needs to be clarified.
  4. The author uses HIV infection as a way to highlight the importance of the immune system, however, the mechanisms are not clear. I suggest that autoimmune disease as risk factors also are exemplified and the mechanisms with HIV clarified if to be included. Other infectious diseases if relevant should be mentioned, ie TB.
  5. In the section of fetal-maternal communication, the author needs to expand on the growing field of microparticles (STBEVs) as introduced by Redman et al. The importance of the endothelial damage occurring in stage 2. These particles do not only active the endothelium, but they also fuse with them, see Cronqvist et al. Furthermore, their content of miRNA can reprogram the endothelium causing long-term consequences. The long-term cardiovascular risk factors should be emphasized better in this respect.
  6. Yet another factor that has emerged during the last decade and that may have important implications on the maternal hemodynamics is free fetal haemoglobin. A recent meta-analysis has shown a significant role for free fetal haemoglobin and scavenger proteins such as hemopexin and alpha-1-microglobulin. The author ought to include this concept as a placental derived factor that alters maternal hemodynamic by I) scavenging NO, II) inducing inflammation III) inducing oxidative stress, IV) causing matrix damage and V) causing direct kidney damage.
  7. THE MATERNAL HEMODYNAMIC SECTION is the most novel part of the review, and it should be expanded and clarified. The cardio-renal syndrome should be better explained, particular in relation to RAS. The different subtypes of PE in relation to cardiorenal communication and venous overflow is hard to follow and the figures do not help. Use fewer panels in figure 3 in favour of enlargement ant clarification. Figure 6 not clear.
  8. The venous compartment hypothesis is beautiful but should tie better into the modified two-stage model.
  9. Do antiangiogenic factors have a different effect on the venous system?

Author Response

General reply:

To meet some comments made by all three reviewers, major changes were made by:  

  • Adding an extra table (Table 2) explaining the sequence of consecutive steps in the pathophysiology of preeclampsia. For readers’ comprehension, this table is presented at the beginning of the pathophysiology chapter and contains specific references to related figures further on.
  • redesigning figure 4 extensively, to illustrate better that the kidney (and renocardial syndrome ) in this figure is used as a proxy for the placenta (and preeclampsia) in pregnant women, as well as for the heart in cardiorenal syndrome [89]. Other syndromes as cardiohepatic and hepatorenal syndrome have also been reported, all reflecting the fundamental pathophysiologic concepts explained in this manuscript: dysfunctional organs trigger a cascade of events eventually leading to a systemic inflammatory response syndrome, initiating dysfunctioning of other distant organs. Table 3 (old Table 2) serves to illustrate this concept. In order to better highlight this message, the title has changed to : “Preeclampsia is a syndrome with a cascade of pathophysiologic events” and the conclusions of abstract and text have been adapted to “…is to be considered a cascade of consecutive events, fundamentally not unique to pregnancy.”  

Replies to comments of reviewer 1

1. Among the less-frequently reported factors, the author list is air pollution. There is actually a growing bulk of reports confirming this association, also on a molecular level. This risk factor should be listed under environmental factors and perhaps a comment made in relation to the increasing incidence.

Reply: thanks for the suggestion. At lines 115-118 is added: “Environmental factors, such as air pollution, are becoming a more and more important predisposing factor for preeclampsia and cardiovascular disease, most likely via oxidative stress induced endothelial dysfunction [21]. This mechanism will be explained further in the chapter on Immune system, implantation, and vascular remodeling.”

2. The two-stage model by Redman is well established. There is an updated version by Redman/Staff that would be more appropriate and necessary to include. This model suggests an explanation for late-onset PE. The author could perhaps join the venous overflow hypothesis with the “pot bound” placenta-if the venous return from the placenta is impaired, this would affect placental function. The problem ought to become more significant as the placenta grows, hence this mechanism is more important in late-onset PE.

Reply: Thank you for citing the new reference, which now replaces the original reference 99 (Staff AC. The two-stage placental model of preeclampsia: An update. J Reprod Immunol. 2019;134-135:1-10. doi:10.1016/j.jri.2019.07.004). The two stage model is documented and illustrated  several times throughout the paper under late oxidative stress in Table 2 and in the chapters on placentation and maternal hemodynamics. The link with venous congestion is presented in Figure 4 and in the discussion of volume overload as a cause of endothelial dysfunction, and is illustrated clinically in the longitudinal observation of Bosio et al [38].

3. Placental hypoxia has been described by Sothil et al. but uneven perfusion of the placenta is likely more important for the oxidative stress. Hypoxia as a factor is controversial, this needs to be clarified.

Reply : At lines 190-195 is written: “Consistent with theoretical models describing the increased flow velocities and oxygen tension in the intervillous space downstream from narrow spiral arteries [46], transabdominal near-infrared spectroscopic measurements of the oxygenation of placental tissue confirmed greater placental oxygenation during PE than during normal pregnancies [47]. This contradicts the concept of placental hypoxia as the primary trigger for the cascade of events that ultimately causes PE.”

4. The author uses HIV infection as a way to highlight the importance of the immune system, however, the mechanisms are not clear. I suggest that autoimmune disease as risk factors also are exemplified and the mechanisms with HIV clarified if to be included. Other infectious diseases if relevant should be mentioned, ie TB.

Reply : In the new Table 2 is now better explained that auto-immune disorders, such as antiphospholipid syndrome, are very strong predisposing factors for preeclampsia [13.14]. On the other hand, immune suppression as during HIV disease lowers the risk for preeclampsia [68]. Maternal infections not interfering with the hosts immune system, such as TB, are not associated with an increased risk for preeclampsia (Loto OM, Awowole I. Tuberculosis in pregnancy: a review. J Pregnancy. 2012;2012:379271. doi:10.1155/2012/379271).

5. In the section of fetal-maternal communication, the author needs to expand on the growing field of microparticles (STBEVs) as introduced by Redman et al. The importance of the endothelial damage occurring in stage 2. These particles do not only active the endothelium, but they also fuse with them, see Cronqvist et al. Furthermore, their content of miRNA can reprogram the endothelium causing long-term consequences. The long-term cardiovascular risk factors should be emphasized better in this respect.

Reply: At lines 218-222 is cited: “These effects are enhanced by epigenetic gene modifications, including altered histone modifications, DNA methylation, and the production of noncoding RNAs, such as microRNAs or circular RNAs [54]. Epigenetic gene modification not only occurs in placental tissue, but also in circulating leucocytes, shed extracellular trophoblast particles, and cell-free DNA and RNA.”

At lines 315-330 is cited under the subheading Syncytiotrophoblast extracellular vesicles : “As mentioned above, an important fetal–maternal communication system involves the intravascular shedding of placental particles, varying in size and shape from multinucleated syncytial aggregates to subcellular nanovesicles, originating from apoptosis in normal pregnancies, but also from necrosis in PE [73]. This phenomenon is also associated with increased serum levels of total cell-free DNA [74]. Both in vitro and in vivo animal studies have shown that these particles are cleared from the circulation via phagocytosis by macrophages and endothelial cells at locations distant from the uterus, where they induce a reactive endothelium-cell response. In uncomplicated pregnancies, endothelial cells become progressively less sensitive to vasoconstrictive mediators. However, in PE, they show signs of activation, including increased surface expression of monocyte adhesion receptors, such as E-selectin, and secretion of proinflammatory IL6 and transforming growth factor β (TGF-β). This activation process spreads rapidly via paracrine and endocrine pathways to other nearby or distant endothelial cells [73]. As discussed above, these particles may act via intravesicular microRNAs and/or circular RNAs that, after phagocytosis by endothelial and immune cells, induce sterile inflammation and alter the production of mediators of angiogenesis and vasoactivity, such as sFLT1, VEGF, and pregnancy-associated placental protein A (PAPP-A) [55,56,75].”

6. Yet another factor that has emerged during the last decade and that may have important implications on the maternal hemodynamics is free fetal haemoglobin. A recent meta-analysis has shown a significant role for free fetal haemoglobin and scavenger proteins such as hemopexin and alpha-1-microglobulin. The author ought to include this concept as a placental derived factor that alters maternal hemodynamic by I) scavenging NO, II) inducing inflammation III) inducing oxidative stress, IV) causing matrix damage and V) causing direct kidney damage.

Reply: Thank you for bringing up this import topic. As presented in Table 2 and Figure 4, local tissue damage is a consequence of oxidative stress, which may present in the early stages of placentation or develop during the course of pregnancy. Fetal hemoglobin is a placental product that is released in the maternal circulation at higher concentrations in preeclampsia than in normal pregnancy. This triggers a cascade of effects contributing to enhanced endothelial dysfunction, systemic inflammatory response and organ damage. As such, this mechanism is part of the transition from local oxidative stress towards a systemic inflammatory response syndrome.  This mechanism is added to Table 3 and to the text at lines 215-217 as “In turn, oxidative stress triggers intravascular inflammation and endothelial dysfunction via the release of factors such as tumor necrosis factor-α (TNFα), interleukin-6 (IL6), IL10, C-reactive protein, and other factors as Hemoglobin F [130], with a new reference: Kalapotharakos G, Murtoniemi K, Åkerström B, et al. Plasma Heme Scavengers Alpha-1-Microglobulin and Hemopexin as Biomarkers in High-Risk Pregnancies. Front Physiol. 2019;10:300. Published 2019 Apr 4. doi:10.3389/fphys.2019.00300

7. THE MATERNAL HEMODYNAMIC SECTION is the most novel part of the review, and it should be expanded and clarified. The cardio-renal syndrome should be better explained, particular in relation to RAS. The different subtypes of PE in relation to cardiorenal communication and venous overflow is hard to follow and the figures do not help. Use fewer panels in figure 3 in favour of enlargement ant clarification. Figure 6 not clear.

Reply: As mentioned above, a new Table 2 is added to the manuscript to guide the reader in linking textual passages with related figures. Figure 3 is a new version of the figures already published in [87] and [98].

8. The venous compartment hypothesis is beautiful but should tie better into the modified two-stage model.

Reply: See reply to comment 2 : The two stage model is documented and illustrated  several times throughout the paper under late oxidative stress in Table 2 and in the chapters on placentation and maternal hemodynamics. The link with venous congestion is presented in Figure 4 and in the discussion of volume overload as a cause of endothelial dysfunction, and is illustrated clinically in the longitudinal observation of Bosio et al [38].

9. Do antiangiogenic factors have a different effect on the venous system?

Reply. This is a very interesting question. At the moment, no vein specific biologicals are known apart from pharmacologic agents as Magnesium Sulphate and Nitrates [87]. As the arterial and venous systems are driven by forces and volumes respectively and serve totally different functions, it is very likely that vein specific mediators will be identified somewhere in the future. 

Reviewer 2 Report

I am pleased to review this well-written review about the pathophysiology of pre-eclampsia. My concerns:

  1. Are the figures original? and if not has the author acquired proper and official copyright rights ?
  2. The author is kindly requested to revise the manuscript as far as phrases such as 'The conclusion from this review is; in line 40, page 2. It reminds of a chapter of a book, and this is a well written review paper.
  3. Last, and not least, the author is kindly requested to conform the manuscript, consulting with SANRA Statement. (https://researchintegrityjournal.biomedcentral.com/articles/10.1186/s41073-019-0064-8) 

Author Response

General reply:

To meet some comments made by all three reviewers, major changes were made by: 

  • Adding an extra table (Table 2) explaining the sequence of consecutive steps in the pathophysiology of preeclampsia. For readers’ comprehension, this table is presented at the beginning of the pathophysiology chapter and contains specific references to related figures further on.
  • redesigning figure 4 extensively, to illustrate better that the kidney (and renocardial syndrome ) in this figure is used as a proxy for the placenta (and preeclampsia) in pregnant women, as well as for the heart in cardiorenal syndrome [89]. Other syndromes as cardiohepatic and hepatorenal syndrome have also been reported, all reflecting the fundamental pathophysiologic concepts explained in this manuscript: dysfunctional organs trigger a cascade of events eventually leading to a systemic inflammatory response syndrome, initiating dysfunctioning of other distant organs. Table 3 (old Table 2) serves to illustrate this concept. In order to better highlight this message, the title has changed to : “Preeclampsia is a syndrome with a cascade of pathophysiologic events” and the conclusions of abstract and text have been adapted to “…is to be considered a cascade of consecutive events, fundamentally not unique to pregnancy.”  

Replies to comments of reviewer 2

  1. Are the figures original? and if not has the author acquired proper and official copyright rights ?

Reply: The figures are original and drawn by a professional medical artist.

  1. The author is kindly requested to revise the manuscript as far as phrases such as 'The conclusion from this review is; in line 40, page 2. It reminds of a chapter of a book, and this is a well written review paper.

Reply: The abstract conclusion has changed to : The helicopter view on pathophysiologic processes associated with preeclampsia, as presented in this paper, illustrates that the etiology of preeclampsia cannot be reduced to one single mechanism, but is to be considered a cascade of consecutive events, fundamentally not unique to pregnancy.  

  1. Last, and not least, the author is kindly requested to conform the manuscript, consulting with SANRA Statement. (https://researchintegrityjournal.biomedcentral.com/articles/10.1186/s41073-019-0064-8) 

Reply: To the introduction section was added: For ages, numerous hypotheses on the etiology and pathophysiology of preeclampsia have been reported, without a generally accepted consensus today. This narrative reviews aims to combine currently published evidence on the sequence of background mechanisms from periconception to the full clinical syndrome preeclampsia, and put these into perspective with the known pathophysiologic processes of systemic syndromes in non-pregnant individuals as cardiorenal syndrome and other. For this, an extended literature search in PubMed was conducted using combinations of the key words: preeclampsia, gestational hypertensive disorders, epidemiology, risk factors, classification, early onset preeclampsia, placental preeclampsia, late onset preeclampsia, maternal preeclampsia, spiral artery, spiral artery remodeling, gestational adaptation, gestational physiology, intervillous space, immune tolerance, inflammation, oxidative stress, systemic inflammatory response syndrome, epigenetics, microRNA, syncytiotrophoblast extracellular vesicles, trophoblast plugs, uterine arteriovenous anastomoses, nitric oxide, maternal hemodynamics, venous hemodynamics, venous congestion, intraabdominal pressure, abdominal hypertension, abdominal compartment syndrome.

The SANRA score sheet is completed as:

  • Justification of the article’s importance for the readership: 2
  • Statement of concrete aims or formulation of questions 2
  • Description of literature search 1
  • Referencing 2
  • Scientific reasoning 2
  • Appropriate presentation of data 2
  • Sumscore:        11                                           

Reviewer 3 Report

In this article the author summarizes the current knowledge on preeclampsia.the author may consider to introduce a figure

Overall this is a very comprehensive manuscript, summarizing the recent Knowledge.

Comments:

  • Introduction might benefit from subheadings
  • Page 6, line 136: the author may consider to introduce / illustrate this Statement (3 subtypes of PE)
  • Placentation process: as the introduction subheadings or dividing the low-text into some paragraphs may help the reader to better follow the text, in the current form it very hard to follow and get the message. Same for the following chapters "Immune System...", "Fetal-maternal communication", "maternal haemodynamics", "venous haemodynamics..."
  • Within the text, there seems to be a Problem with occurence of figures. It starts with fig 2 on page 8. I assume that this is figure 1 on page 33 and is related to the text starting on page 7, line 162/163?
  • Page 8, line 183 and 184: please exchange the wording "see below" by introducing the correct chapters/headings (maybe also in line 195).
  • I suggest to think about introducing a figure for better illustration of the text/message on page 8 (mainly ox. stress)
  • Continuing the figure "problem" - on page 9 figure 1c is introduced. Is this figure 2 (page 35) and actually there "line" 4? If so, where to find in the figure (as written on line 208/209) ""Diversion of the arterial blood...by AV anastomoses". Looking at figure on page 33, it seems as the "lower panel" shows AV that this might be 1C
  • Page 10, line 232: 1B - to which figure is this related?
  • Same on page 11, line 258 - Fig 1A - which figure actually?
  • Illustration for text on page 12 would be of help
  • page 13, line 324: again figure 2 - is this the same as the previous figure "2"?
  • Isn't "venous haemodynamcis" a part of maternal haemodynamics?

Author Response

General reply:

To meet some comments made by all three reviewers, major changes were made by: 

  • Adding an extra table (Table 2) explaining the sequence of consecutive steps in the pathophysiology of preeclampsia. For readers’ comprehension, this table is presented at the beginning of the pathophysiology chapter and contains specific references to related figures further on.
  • redesigning figure 4 extensively, to illustrate better that the kidney (and renocardial syndrome ) in this figure is used as a proxy for the placenta (and preeclampsia) in pregnant women, as well as for the heart in cardiorenal syndrome [89]. Other syndromes as cardiohepatic and hepatorenal syndrome have also been reported, all reflecting the fundamental pathophysiologic concepts explained in this manuscript: dysfunctional organs trigger a cascade of events eventually leading to a systemic inflammatory response syndrome, initiating dysfunctioning of other distant organs. Table 3 (old Table 2) serves to illustrate this concept. In order to better highlight this message, the title has changed to : “Preeclampsia is a syndrome with a cascade of pathophysiologic events” and the conclusions of abstract and text have been adapted to “…is to be considered a cascade of consecutive events, fundamentally not unique to pregnancy.”  

Replies to comments of reviewer 3

  • Introduction might benefit from subheadings

Reply : Subheadings have been added

  • Page 6, line 136: the author may consider to introduce / illustrate this Statement (3 subtypes of PE)

Reply : At line 135-136 is cited: “The first classification system discriminated early-onset from late-onset PE based on whether the clinical diagnosis was made before or after 34 weeks [27].”

At lines 153-159 is cited: “To complete the picture, two different subtypes of late-onset PE have also been reported, based on the bimodal skewing of the birth weight percentiles and uterine artery Doppler ultrasound measurements [35]. The categorization of PE into three subtypes is not only consistent with the reported prevalence figures, which are higher for late-onset PE than for early-onset PE [36], but also with the three reported types of longitudinal changes in maternal cardiovascular functions throughout the course of the pregnancy, as will be explained below [37–39].”

  • Placentation process: as the introduction subheadings or dividing the low-text into some paragraphs may help the reader to better follow the text, in the current form it very hard to follow and get the message. Same for the following chapters "Immune System...", "Fetal-maternal communication", "maternal haemodynamics", "venous haemodynamics..."

Reply: subheadings have been added

  • Within the text, there seems to be a Problem with occurence of figures. It starts with fig 2 on page 8. I assume that this is figure 1 on page 33 and is related to the text starting on page 7, line 162/163?

Reply : An extra table (Table 2) is added, explaining the sequence of consecutive steps in the pathophysiology of preeclampsia. For readers’ comprehension, this table is presented at the beginning of the pathophysiology chapter and contains specific references to related figures further on.

  • Page 8, line 183 and 184: please exchange the wording "see below" by introducing the correct chapters/headings (maybe also in line 195).

Reply: Referencing the appropriate chapters has been performed throughout the manuscript

  • I suggest to think about introducing a figure for better illustration of the text/message on page 8 (mainly ox. stress)

Reply: Figure 4 has been redesigned to better explain the role and mechanisms of oxidative stress in the development of a systemic inflammatory response syndrome

  • Continuing the figure "problem" - on page 9 figure 1c is introduced. Is this figure 2 (page 35) and actually there "line" 4? If so, where to find in the figure (as written on line 208/209) ""Diversion of the arterial blood...by AV anastomoses". Looking at figure on page 33, it seems as the "lower panel" shows AV that this might be 1C. Page 10, line 232: 1B - to which figure is this related? Same on page 11, line 258 - Fig 1A - which figure actually? Illustration for text on page 12 would be of help. page 13, line 324: again figure 2 - is this the same as the previous figure "2"?

Reply : An extra table (Table 2) is added, explaining the sequence of consecutive steps in the pathophysiology of preeclampsia. For readers’ comprehension, this table is presented at the beginning of the pathophysiology chapter and contains specific references to related figures further on.

  • Isn't "venous haemodynamcis" a part of maternal haemodynamics?

Reply: the chapter venous hemodynamics is now presented under subheading as a part of the chapter “maternal hemodynamics”

Round 2

Reviewer 2 Report

No further comments.